# Unexpected Room Temperature Ferromagnetism of a Ball-Milled Graphene Oxide—Melamine Mixture

**DOI:** 10.3390/molecules27227698

**Published:** 2022-11-09

**Authors:** Vladimir P. Vasiliev, Eugene N. Kabachkov, Alexander V. Kulikov, Roman A. Manzhos, Iurii G. Morozov, Yury M. Shulga

**Affiliations:** 1Federal Research Center of Problems of Chemical Physics and Medicinal Chemistry of RAS, Acad. Semenov Ave., 1, 142432 Chernogolovka, Russia; 2Merzhanov Institute of Structural Macrokinetics and Materials Science of RAS, Acad. Osipyan St., 8, 142432 Chernogolovka, Russia; 3Department of Functional Polymer Materials, National University of Science and Technology MISIS, Leninsky Ave., 4, 119049 Moscow, Russia

**Keywords:** graphene oxide, melamine, ball-milling, electron spin resonance, room temperature ferromagnetism

## Abstract

Nitrogen-doped carbon nanomaterial (NDCNM) was synthesized by grinding a mixture of graphene oxide and melamine in a planetary mill with both balls and milling chamber of zirconium dioxide. In the electron spin resonance spectrum of NDCNM, a broad signal with *g* = 2.08 was observed in addition to a narrow signal at *g* = 2.0034. In the study using a vibrating-sample magnetometer, the synthesized material is presumably a ferromagnet with a coercive force of 100 Oe. The specific magnetization at 10,000 Oe is approximately 0.020 and 0.055 emu/g at room temperature and liquid nitrogen temperature, respectively.

## 1. Introduction

Organic ferromagnetism was first loudly announced in Ref. [1]. It reported the ferromagnetism of the product of polymerization of paramagnetic stable biradical, 1,4-bis-(2,2,6,6-tetramethyl-4-oxy-4-piperidyl-1-oxyl)-butadiin, (BIPO) and described previous experimental and theoretical work on the designated topics. In particular, the theoretical prediction of 1D organic ferromagnets was noted to be proposed by Ovchinnikov in 1978 [2].

The next bright page in this story is the work of [3], which was published in 2001. The authors wrote that as a result of C_60_ polymerization under high pressure and high temperature, a material is formed that exhibits features characteristic of ferromagnets: “saturation magnetization, large hysteresis and attachment to a magnet at room temperature”. However, 5 years later the authors published a communication in which they doubt their results [4]. This dramatic publication is not often cited. Yet, we believe that the problem raised by this offers very instructive experience.

At present, quite a lot of publications are devoted to the ferromagnetism of *d*^0^ materials (see, for example, [5,6,7,8,9,10,11,12,13,14,15,16,17,18,19,20]). There are also many works about the ferromagnetism of graphene-like structures [21,22,23,24,25,26,27]. The situation with carbon-based materials seems to us to be very interesting. For example, ferromagnetism has been observed in pure carbon structures such as the already mentioned rhombohedral C_60_ fullerite [3], highly oriented pyrolytic graphite [28,29], glassy carbon after laser ablation [30], and double-layer graphene [31]. Ferromagnetism was discovered for hydrogenated C_60_ fullerite [32] and graphene [33]. There are also numerous publications where ferromagnetism has been found in graphene oxide, reduced graphene oxide, and heteroatom doped GO and/or rGO [34,35,36,37,38,39,40,41,42,43,44,45,46,47,48]. One might assume that the issue was resolved unambiguously in favor of the existence of ferromagnetic organic and carbon materials. However, there are experimental works which prove that paramagnetic defects in graphene structures, even at their high concentration, are still far enough apart for an (anti)ferromagnetic exchange to occur between them. Of particular note is the work of Geim et al., who studied the effect of fluorination and irradiation with hydrogen and carbon atoms on the magnetic properties of graphene [49]. Based on the results of their study, the authors concluded that defective ferromagnetism is impossible in graphene structures. The absence of ferromagnetic or superparamagnetic contribution to the magnetization of graphene oxide measured up to 5 K was also stated in [50].

A simple method of production of an efficient platinum-free oxygen reduction reaction electrocatalyst was described in [51,52]. The electrocatalyst was synthesized by a solid-phase method as a result of grinding graphene oxide and melamine in a planetary ball-mill machine without the use of any solvents or high-temperature processing. Based on the XPS and IR spectroscopy data, high electrocatalytic activity of the obtained material was assumed to be determined by the presence of nitrogen atoms and quinone groups on its surface.

In this study we present the results of the analysis of obtained material by electron paramagnetic resonance (ESR) and magnetometry methods. Surprisingly, the ESR spectrum contains a fairly intense broad signal with a *g* factor of 2.08 in addition to the narrow ESR signal with a *g* factor of 2.0034, which is characteristic of materials based on graphene oxide. A vibrating-sample magnetometry study showed that the resulting material is presumably a ferromagnet with a specific saturation magnetization of 0.02 emu/g at room temperature. Elemental analysis, IR and X-ray photoelectron spectroscopy were used to characterize the samples.

## 2. Experimental

### 2.1. Synthesis of Nitrogen-Doped Carbon Material

Graphene oxide was synthesized using a modified Hummers’ method [53]. Melamine, C_3_N_6_H_6_ (99.9%, BASF SE (Ludwigshafen am Rhein, Germany)) was used as a source of nitrogen.

The mechanochemical synthesis was carried out in a «FRITSCH pulverisette 6» planetary mill. Grinding vessel and balls were of ZrO_2_, the internal diameter of the grinding vessel, the volume, and the ball diameter were 65 mm, 85 mL, and 10 mm, respectively. The GO/melamine ratio was 4:1, rotation speed was 400 rpm and grinding time was 6 min. After grinding, the resulting powder was kept for 1 h in a 10% aqueous solution of ammonia, treated in an ultrasonic bath, and then it was centrifuged and washed with water 4–5 times to remove melamine residues [52].

### 2.2. Characterization

Elemental analysis of the samples preliminarily degassed in an argon flow at a temperature of 80 °C for 30 min was carried out on a Vario Micro cube CHNS analyzer (Elementar GmbH, Hanau, Germany).

The IR spectra were recorded at room temperature in the range of 400–4000 cm^−1^ on a Perkin-Elmer “Spectrum Two” Fourier-transform spectrometer (Waltham, MA, USA) with an ATR attachment with a diamond crystal.

The ESR spectra of the powders were recorded with a Bruker Elexsys II E 500 ESR spectrometer and an SE/X 2544 radio spectrometer (Radiopan, Poznan, Poland) at room temperature. The number of spins *N* and the *g* factor were determined using the Xepr software. To check the correctness of these procedures, a weighed sample of CuSO_4_·5H_2_O and DPPH with a *g* factor of 2.0036 were used. The accuracy of concentration determination was ca. 15%.

Magnetic characteristics were measured using an M4500 vibrating-sample magnetometer (EG&G PARC, Gaithersburg, MD, USA) in magnetic fields up to 10 kOe at room temperature and liquid nitrogen temperature. The diamagnetic signal of the nylon sample holder was subtracted from the obtained total magnetization of material.

XPS spectra were obtained using a Specs PHOIBOS 150 MCD electronic spectrometer for chemical analysis (SPECS GmbH, Berlin, Germany). During the measurement of spectra, the vacuum in the spectrometer chamber did not exceed 2 × 10^−10^ Torr; the X-ray tube was equipped with a magnesium anode (Mg Kα radiation is 1253.6 eV) and the source power was 225 W. The survey spectrum was recorded in the range of 0–1000 eV in the constant transmission energy mode (40 eV for the survey spectrum and 10 eV for individual lines). The survey spectrum was recorded with a step of 1.0 eV, while the spectra of individual lines with a step of 0.05 eV.

## 3. Results

### 3.1. Elemental Analysis and SEM

Table 1 presents the data on the composition of graphene oxide (before and after grinding), melamine (before and after grinding), and grinding products of two GO:melamine mixtures after removal of unreacted melamine. As is seen, the grinding does not very strongly affect pure GO and melamine. However, the composition of a grinding product of the GO:melamine mixture is not a simple sum of the initial components. Doping of graphene oxide with nitrogen can be assumed to occur as a result of grinding. Since the product mainly contains carbon, hereinafter we will refer to it as nitrogen-doped carbon nanomaterial (NDCNM). We suggest that presence of sulfur in samples is due to the GO production by Hummers’ method. The intense washing of graphite oxide by water, probably, is not sufficient to completely remove traces of sulfuric acid, which might be trapped in closed pores (see [54]).

The SEM micrographs of graphene oxides are well known. The GO micrograph obtained by us (Figure 1) resembles those given in [55,56,57]. The surface of the GO sample consists of smooth sheets that have curves and folds. One can also distinguish the edges of individual sheets. The surface of the NDCNM sample is formed of rounded particles, the dimensions of which are generally less than 100 nm. We failed to notice the morphological features typical for GO sheets in the NDCNM micrograph.

### 3.2. IR Spectra

IR spectra of graphene oxide, melamine, and NDCNM—the product of mechanochemical treatment of their 4:1 mixture—is presented in Figure 2. Comparing the spectra, one can note that the IR spectrum of NDCNM differs from the spectra of initial components. In the NDCNM spectrum, there are no absorption bands of stretching vibrations of N–H bonds, which can be observed in the spectrum of pure melamine powder at 3468, 3417, 3324 and 3121 cm^−1^ [58]. The absence of these peaks can be interpreted as dissociation of N–H bonds as well as the complete removal of amine groups as a result of grinding. At the same time, the appearance of a number of absorption bands in the NDCNM spectrum in the region from 2350 to 1900 cm^−1^ can be ascribed to the presence of cyano groups in various configurations in the sample [59,60]. Note, IR spectra were obtained in vacuum, so we exclude the manifestation of vibrations of CO_2_ molecules of the gaseous phase in this region. Finally, in our opinion, the wide absorption band in the region from 3700 to 3000 cm^−1^ in the GO and NDCNM spectra is mainly due to the stretching vibrations of O–H groups bound by hydrogen bonds, which determines such a wide absorption band [61]. However, the full width at half maximum of this band in the NDCNM spectrum is less than that in the GO spectrum, and this band is shifted towards higher wavenumbers (3333 cm^−1^ for NDCNM and 3173 cm^−1^ for GO). This means that hydrogen bonds in NDCNM are weaker than the hydrogen bonds in graphene oxide. It is well known that strengthening of hydrogen bonds shifts the frequency of O−H vibrations to low frequencies by hundreds of reciprocal centimeters [62].

Comparing the spectrum of NDCNM with the spectrum of GO, one can see that the absorption band at 1730 cm^−1^ due to the stretching vibrations of C=O bonds [63] is practically absent in the NDCNM spectrum.

### 3.3. XPS Spectra

The elemental composition of some samples was also calculated using analytical lines of the survey XPS spectrum (Figure 3). As is seen, the content of oxygen in the layer analyzed by XPS is substantially less for NDCNM than for GO sample. The carbon content in the NDCNM is slightly higher (by 2.4 at.%) than that in initial GO; that is, graphene oxide is slightly reduced as a result of grinding (Table 2). As expected, the content of nitrogen in NDCNM sample is rather high. It should be mentioned that even during long-term data acquisition, no peaks of the iron group elements (Fe, Co, Ni) appeared in the XPS spectra.

The identification of surface nitrogen-containing groups can be carried out on the basis of an analysis of the fine structure of the N1s line in the spectrum. According to the literature (see [64] and references therein), pyridinic nitrogen (N1) appears in the range of 398.0–399.3 eV and pyrrolic nitrogen (N2) appears in the range of 399.8–401.2 eV) in the XPS spectra of nitrogen-doped carbon materials. Of note, the N1s lines of amino (399.1 eV [64]) and cyano (399.3 eV [64]) groups are also located in this region. Therefore, it is difficult to clearly identify them. The peak corresponding to the nitrogen atoms of the N4 type (inside a graphite sheet) is located at about 401 eV, and the peak corresponding to terminal graphite nitrogen (N3) is at 402.3 eV. Oxidized pyridine nitrogen (N5) corresponds to a peak at 402.8 eV. Physically adsorbed nitrogen (N5) corresponds to a peak at 404.7 eV [64].

Three peaks can be distinguished in the N1s spectrum of NDCNM (Figure 4a). Unexpectedly, the major contribution to the N1s line can be assigned to the pyrrolic nitrogen (N2).

C1s spectrum of NDCNM (Figure 4b) is also deconvoluted into four peaks at 284.7 (peak 1), 286.8 (peak 2), 288.7 eV (peak 3) and 290.6 eV (peak 4). These peaks can be assigned to carbon atoms not bonded with oxygen atoms (peak 1), carbon atoms singly bonded with an oxygen atom (peak 2), and carbon atoms which have two bonds with one or two oxygen atoms (peak 3) [65]. According to [66], the concentration of C(sp^3^) in NDCNM is 30–35%.

### 3.4. ESR Spectra

ESR spectra of initial GO and GO ground in a planetary mill are presented in Figure 5. As is seen, one narrow signal is observed in the ESR spectra of the samples at room temperature. At the same time, a narrow peak with *g* = 2.0034 is present in the NDCNM spectrum along with a broad line with *g* = 2.08. The nature of the narrow ESR signal for graphene oxide samples has been studied in many works. At present, this signal is considered to be due to paramagnetic centers associated with isolated carbon atoms in highly functionalized regions of graphene oxide, as well as to delocalized π electrons of aromatic domains [67]. The signal from paramagnetic centers of the first type corresponds to *g* > 2.003 and a half-width of 2.5 Oe; the signal from π electrons is wider (up to 12 Oe) with *g* = 2.002. The obtained ESR spectra of initial and ground GO indicate that localized paramagnetic centers predominate in the sample of initial GO, while milling causes a shift towards delocalized states, which is primarily evidenced by an increase in the signal half-width. In the case of NDCNM, the half-width of the narrow signal is also greater than that for initial GO. The nature of the broad signal has not yet been described in the literature.

### 3.5. Magnetic Measurements

σ(H) curves were measured at two temperatures (78 and 296 K) with a change in the magnetic field from −10 to 10 kOe at a constant rate of 1.2 kOe/min. The NDCNM sample showed σ(H) dependence with a well-resolved hysteresis loop typical of a ferromagnetic sample (Figure 6). The coercive force was about 100 Oe. Within the experimental error, as the temperature was increased from 78 to 296 K, the magnetization at 10 kOe decreased from ca. 0.05 to ca. 0.02 emu/g. This indicates that the Curie temperature is slightly above 300 K.

## 4. Discussion

In each experimental work dealing with ferromagnets with a low saturation magnetization, it is necessary to keep in mind the previous findings, which make it possible to separate impurity ferromagnetism from internal ferromagnetism. Foremost, for experimental work it is necessary to analyze the studied materials for the traces of the iron group elements (Fe, Co, Ni). Moreover, to estimate the level of impurity ferromagnetism, it is necessary to use the values of saturation magnetization σ_s_ equal to 2.2, 1.7, and 0.6 μ_B_/atom for Fe, Co, and Ni, respectively [32]. It is desirable to check all initial materials for the presence of ferromagnetic impurities to avoid the use of magnetic stirrers, iron tweezers and spatulas and rerun experiments with unexpected results. Note that the sources of ferromagnetic impurities in the study of lightweight samples with a weak magnetic signal by such methods as vibrating-sample magnetometers (VSM) and superconducting quantum interference devices (SQUID) are well described in [8].

As for the data provided in the present study, it seems to us that impurity ferromagnetism cannot explain the obtained results. The materials were ground in a mill, all parts of which were made of non-magnetic zirconium dioxide. Moreover, grinding the graphene oxide and melamine separately did not lead to the formation of ferromagnetic substances. Ferromagnetic structures were obtained only in the case of milling a mixture of graphene oxide and melamine. One of the ferromagnetic samples was carefully analyzed by XPS and no impurities of iron group elements were found in it. The synthesis was repeated four times with slight changes in the procedure of washing the ground sample from unreacted melamine. The results on magnetic properties were reproduced with high accuracy.

Potassium permanganate is known to be often used in the oxidation of graphite [53]. The removal of manganese residues from graphene oxide is also known to be a rather laborious operation. In principle, the ferromagnetism of some manganese compounds with perovskite structure is known (see, for example, [68]). However, it is difficult to imagine that the presence of melamine contributes to the formation of a perovskite structure. We believe that the presence of a small impurity of manganese cannot explain the ferromagnetism observed by us in the product of milling a mixture of graphene oxide and melamine.

## 5. Conclusions

In this research, we report a simple method for producing nitrogen-doped carbon nanomaterial (NDCNM). Elemental analysis, IR, XPS, EPR, and magnetometry were used to characterize the material. Furthermore, along with the narrow ESR signal characteristic of materials based on graphene oxide, an intense broad signal with a *g* factor of 2.08 was detected in the ESR spectrum of the NDCNM sample. A study using a vibrating-sample magnetometer showed that, in terms of its magnetic behavior, the NDCNM can be considered with a high degree of probability to be a weak ferromagnet with a coercive force of 100 Oe and specific magnetization at 10,000 Oe, approximately 0.020 and 0.055 emu/g at room temperature and liquid nitrogen temperature, respectively.

Summarizing the available literature data on this topic, we can conclude that a huge amount of experimental work has been carried out to identify ferromagnetism in various carbon materials. Various theoretical models have been constructed to explain this phenomenon. However, despite this, the appearance of ferromagnetism in carbon materials can still be considered unexpected.

## Figures and Tables

**Figure 1 molecules-27-07698-f001:**
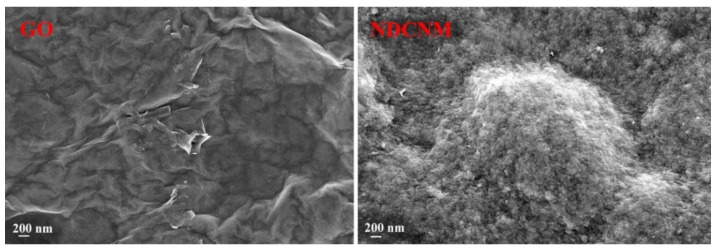
SEM micrographs of GO, and NDCNM.

**Figure 2 molecules-27-07698-f002:**
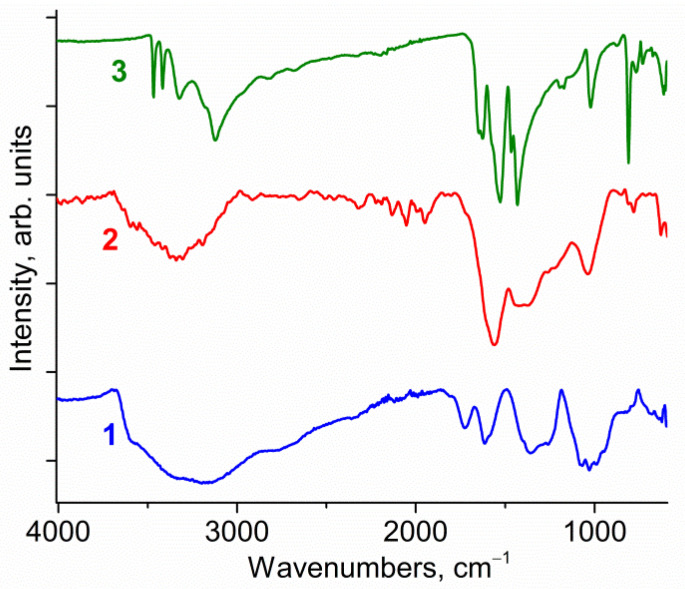
IR spectra of graphene oxide (**1**), NDCNM (**2**) and melamine (**3**).

**Figure 3 molecules-27-07698-f003:**
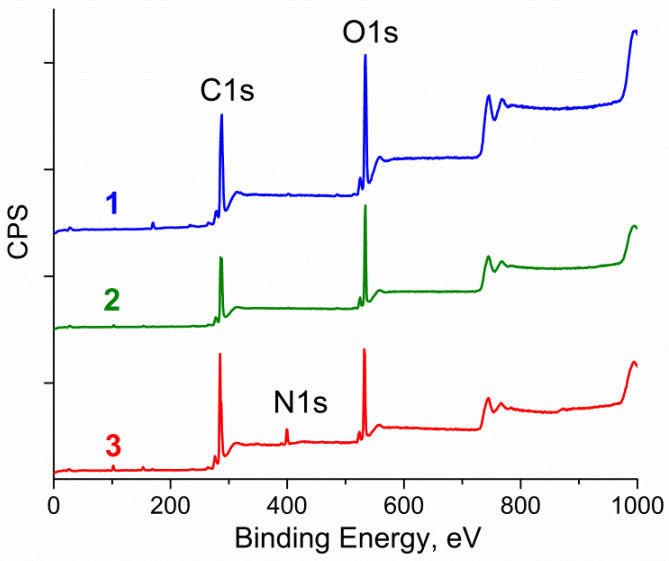
Survey XPS spectra of initial GO (**1**), sample of GO ground in a planetary mill (**2**) and NDCNM powder (**3**).

**Figure 4 molecules-27-07698-f004:**
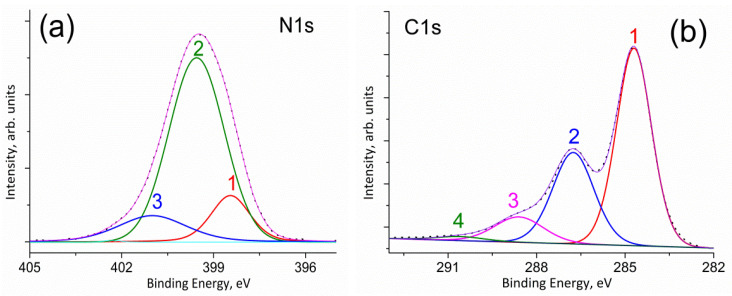
N1s (**a**) and C1s (**b**) high resolution spectra of NDCNM.

**Figure 5 molecules-27-07698-f005:**
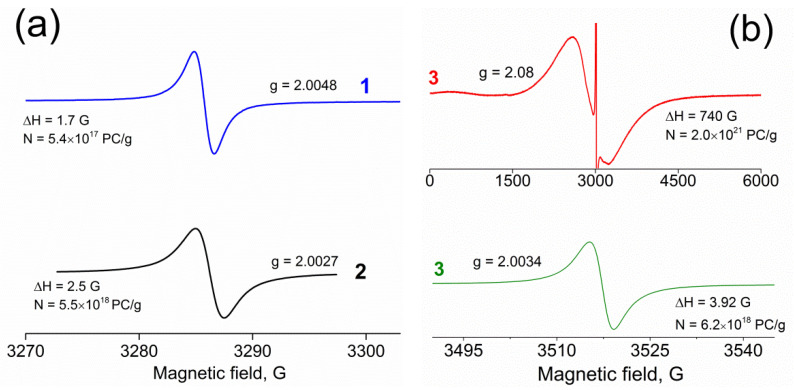
(**a**)—ESR spectra of initial GO (1) and ground GO (2); (**b**)—ESR spectra of NDCNM powder (3) (top, in a wide range of magnetic field; bottom, in a narrow range of magnetic field). Spectra were obtained at a room temperature.

**Figure 6 molecules-27-07698-f006:**
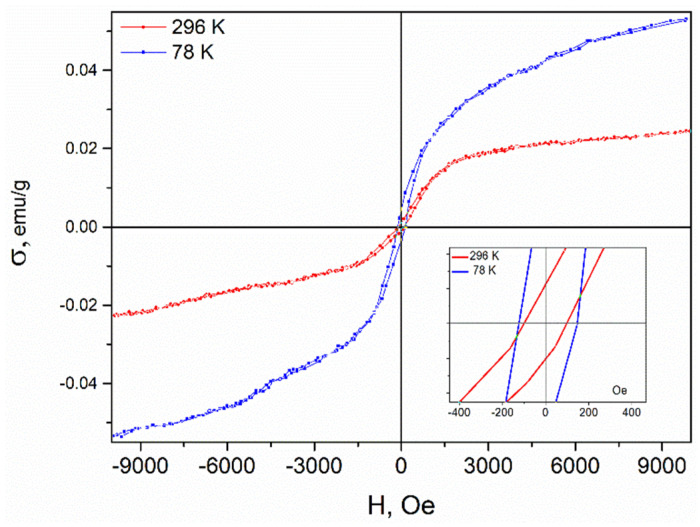
Specific magnetization σ (emu/g) as a function of the magnetic field H (Oe).

**Table 1 molecules-27-07698-t001:** Elemental composition of the studied samples (wt.%).

Sample	C	H	N	S
Melamine *	28.57	4.76	66.67	0.000
melamine (grinding)	28.71	4.544	65.24	0.042
graphene oxide	52.13	2.113	1.07	0.070
GO (grinding)	53.28	2.277	1.13	0.065
NDCNM	52.47	2.473	6.58	0.033

*—calculated according to the chemical formula of melamine (C_3_N_6_H_6_).

**Table 2 molecules-27-07698-t002:** Elemental composition (at.%) of the samples as determined from XPS spectra.

Sample	Element			
	C	N	O	S
graphene oxide	74.3	0.3	23.5	1.9
GO (grinding)	76.8	0.0	23.0	0.2
NDCNM	76.7	5.5	17.4	0.4

## Data Availability

Not applicable.

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
