# Peer review of "Unexpected Room Temperature Ferromagnetism of a Ball-Milled Graphene Oxide—Melamine Mixture"

_molecules, 2022, doi:10.3390/molecules27227698_

Round 1
Reviewer 1 Report
Please refer to the attachment for more details

Author Response
Reply to Reviewer #1 Comments
Vasiliev et al. reported that a broad signal with g = 2.08 was observed in electron paramagnetic resonance (EPR) spectrum of nitrogenous compound nanomaterial mixed with GO and melamine by grinding method, and its elementary composition, IR spectra, XPS spectra, EPR spectra, and magnetic features were also characterized with necessary discussions and analysis. I think the manuscript could be considered for publication after appropriate revisions, and my suggestions were appended underneath. As such, I cannot recommend it for publication in the present form.
Reviewer #1:
1) Claimed that ”the content of oxygen in the layer analyzed by XPS is substantially less for NDCNM than for GO sample. The carbon content in the NDCNM and GO (grinding) samples is slightly higher (by 2.4-2.5 at.%) than that in initial GO; that is, graphene oxide is slightly reduced as a result of grinding” in Fig.2, please explain the reasons for this discrepancy.
Authors reply:
There was indeed not accurate phrase in the manuscript. Here we wanted to say that the graphene oxide in the milled mixture was slightly reduced during grinding. This moment has been clarified in the revised version of manuscript.
Reviewer #1:
2) Claimed that “a clearly pronounced hysteresis and a coercive force of 100Oe” in Fig. 5, please provide more details about it (e.g. measuring conditions at which temperature, differences with temperature varying and reasons), and re-plot the figure with more clear details as an inset.
Authors reply:
The figure has been modified in accordance with the Reviewer’s comments. Corresponding information was added in the Experimental section.
Reviewer #1:
3) No background, motivation and purpose in the introduction and too much in the conclusion. It seems that the order of the MS is reversed. The MS with more information in introduction while list the collection of bullet points of major findings as conclusions must be re-written and re-framed.
Authors reply:
We agree with the Reviewer’s comment. In accordance with this remark, the introduction has been significantly extend in the revised version of manuscript and conclusions have been formulated more clearly.
Thanks to the Reviewer for helpful comments!

Reviewer 2 Report
The paper is an interesting set of results from what appears to be an experiment with a different original purpose in mind. The writing of the paper clearly demonstrates this fact. I am not sure if it was necessary to structure the paper that way. I have the following comments on the paper.
Major comments
1) The introduction is only two paragraphs and provides very little information about the research desrcibed. Please add more information as to the justification of the experiment and why the studies described were performed.
2) It is stated in the introduction that the NDCNM powder was made specifically as an electrocatalyst for the Oxygen Reduction Reaction. Has it been examined for its performance in this regard? The inclusion of a reference to the performed study would be sufficient.
3) Please include appropriate references for the assigned IR peaks in the experimental results section.
4) In the VSM results it is not readily simple to see ferromagnetic behavior. Can you include an insert with a zoomed in central part of the graph so as to better demonstrate the hysteresis.
5) You demonstrate possible effects of ferromagnetic contamination on the produced material. Would it be possible to look for Iron, Nickle or Cobalt in the elemental analysis step? If so include these results.
Minor comments
1) It's the "Hummer's method" not "hammers" please correct.
2) Figure 4's labels are not in english. Please edit to make it consistent with the rest of the manuscript.
3) Include images, TEM, SEM of the samples if possible.
Author Response
Reply to Reviewer #2 Comments
The paper is an interesting set of results from what appears to be an experiment with a different original purpose in mind. The writing of the paper clearly demonstrates this fact. I am not sure if it was necessary to structure the paper that way. I have the following comments on the paper.
Major comments
Reviewer #2:
1) The introduction is only two paragraphs and provides very little information about the research described. Please add more information as to the justification of the experiment and why the studies described were performed.
Authors reply:
The Introduction and Conclusion sections have been rewritten in the revised version of manuscript.
Reviewer #2:
2) It is stated in the introduction that the NDCNM powder was made specifically as an electrocatalyst for the Oxygen Reduction Reaction. Has it been examined for its performance in this regard? The inclusion of a reference to the performed study would be sufficient.
Authors reply:
The results of the study of electrocatalytic activity of composite are presented in [Vasiliev, V.P.; Manzhos, R.A.; Krivenko, A.G.; Kabachkov, E.N.; Shulga, Y.M. Nitrogen-enriched carbon powder prepared by ball-milling of graphene oxide with melamine: An efficient electrocatalyst for oxygen reduction reaction. Mendeleev Commun. 2021, 31, 529–531]. See ref. [51] in the revised version of manuscript.
Reviewer #2:
3) Please include appropriate references for the assigned IR peaks in the experimental results section.
Authors reply:
We thank Reviewer for the comments. We have added several references with the assignment of peaks in the IR spectra.
Reviewer #2:
4) In the VSM results it is not readily simple to see ferromagnetic behavior. Can you include an insert with a zoomed in central part of the graph so as to better demonstrate the hysteresis.
Authors reply:
To better demonstrate the hysteresis, an insert with a zoomed central part of the graph was added in the Fig. 6 of the revised version of manuscript.
Reviewer #2:
5) You demonstrate possible effects of ferromagnetic contamination on the produced material. Would it be possible to look for Iron, Nickle or Cobalt in the elemental analysis step? If so include these results.
Authors reply:
The remark is clear to us. At present, we have only the results of elemental analysis obtained by the XPS method. They are included in the text of the manuscript.
Minor comments
Reviewer #2:
1) It's the "Hummer's method" not "hammers" please correct.
Authors reply:
We believe, it should be corrected to «Hummers’ method» (see ref. Hummers, W.S.; Offeman, R.E. Preparation of Graphitic Oxide. JACS, 1958, 80, 1339‒1339).
Reviewer #2:
2) Figure 4's labels are not in english. Please edit to make it consistent with the rest of the manuscript.
Authors reply:
We thank Reviewer for the comment. The error was corrected in the revised version of manuscript.
Reviewer #2:
3) Include images, TEM, SEM of the samples if possible.
Authors reply:
SEM images of the samples are introduced in the revised version of manuscript (Fig.1).
Thanks to the Reviewer for helpful comments.
Reviewer 3 Report
The present manuscript by Vladimir P. Vasiliev et al. entitled “Unexpected room temperature ferromagnetism of a ball-milled graphene oxide-melamine mixture” reports on the synthesis and magnetic properties of the ball-milled graphene oxide-melamine mixture. The authors investigated the resulting mixture by several methods like elemental analysis, IR and XPS spectroscopy. Magnetism was studied by magnetometry. In my opinion, the paper cannot be published in its present state due to the following reasons:
- The manuscript was not prepared well. The introduction is very short, missing justification for this work. The quality of some figures is low, e.g. EPR figures still contain Cyrillic letters.
- The magnetization measurements are very noisy and not convincing this referee about ferromagnetism. Considering that the ferromagnetism of the resulting material is the highlight of the present article, this is the major issue. Are there any other measurements that could undoubtedly confirm ferromagnetic ordering? E.g., ZFC/FC or AC measurements?
Author Response
Reply to Reviewer #3 Comments
The present manuscript by Vladimir P. Vasiliev et al. entitled “Unexpected room temperature ferromagnetism of a ball-milled graphene oxide-melamine mixture” reports on the synthesis and magnetic properties of the ball-milled graphene oxide-melamine mixture. The authors investigated the resulting mixture by several methods like elemental analysis, IR and XPS spectroscopy. Magnetism was studied by magnetometry. In my opinion, the paper cannot be published in its present state due to the following reasons:
Reviewer #3:
- The manuscript was not prepared well. The introduction is very short, missing justification for this work. The quality of some figures is low, e.g. EPR figures still contain Cyrillic letters.
Authors reply:
We thank Reviewer for this comment. In full accordance with the Reviewer’s recommendations, the introduction has been extended; the quality of figures has been improved in the revised version of manuscript.
Reviewer #3:
- The magnetization measurements are very noisy and not convincing this referee about ferromagnetism. Considering that the ferromagnetism of the resulting material is the highlight of the present article, this is the major issue. Are there any other measurements that could undoubtedly confirm ferromagnetic ordering? E.g., ZFC/FC or AC measurements?
Authors reply:
The figure with a lower noise level is presented in the revised version of manuscript. In addition to measurements using a vibrating-sample magnetometer, the presence of a broad line in the EPR spectrum, as well as a sharp increase in the intensity of the EPR signal, testifies that the obtained NDCNM sample can be considered with a high degree of probability as a weak ferromagnet.
Due to the extremely short period of time given for the revision, we cannot measure FC (field cooling) и ZFC (zero-field cooling); this study would be performed in the future in order to continue our work.
Thanks to the Reviewer for helpful comments!
Round 2
Reviewer 1 Report
All issues have been addressed, publish as is
Reviewer 3 Report
I am satisfied with the changes the authors made. Nevertheless, I recommend to the authors to carefully check grammar in the manuscript before publishing.